# MeVer NetworkX: Network Analysis and Visualization for Tracing Disinformation

Olga Papadopoulou [1,*,†], Themistoklis Makedas [1,†], Lazaros Apostolidis [1], Francesco Poldi [2], Symeon Papadopoulos [1] and Ioannis Kompatsiaris [1]

1. Centre for Research and Technology Hellas—CERTH, Information Technologies Institute—ITI, 6th km Harilaou-Thermi, Thermi, 57001 Thessaloniki, Greece; tmakedas@iti.gr (T.M.); laaposto@iti.gr (L.A.); papadop@iti.gr (S.P.); ikom@iti.gr (I.K.)
2. EU DisinfoLab, Chaussée de Charleroi 79, 1060 Brussels, Belgium; fp@disinfo.eu
* Correspondence: olgapapa@iti.gr; Tel.: +30-2311-257766
† These authors contributed equally to this work.

**Abstract:** The proliferation of online news, especially during the "infodemic" that emerged along with the COVID-19 pandemic, has rapidly increased the risk of and, more importantly, the volume of online misinformation. Online Social Networks (OSNs), such as Facebook, Twitter, and YouTube, serve as fertile ground for disseminating misinformation, making the need for tools for analyzing the social web and gaining insights into communities that drive misinformation online vital. We introduce the MeVer NetworkX analysis and visualization tool, which helps users delve into social media conversations, helps users gain insights about how information propagates, and provides intuition about communities formed via interactions. The contributions of our tool lie in easy navigation through a multitude of features that provide helpful insights about the account behaviors and information propagation, provide the support of Twitter, Facebook, and Telegram graphs, and provide the modularity to integrate more platforms. The tool also provides features that highlight suspicious accounts in a graph that a user should investigate further. We collected four Twitter datasets related to COVID-19 disinformation to present the tool's functionalities and evaluate its effectiveness.

**Keywords:** social network analysis; network visualization tools; online disinformation; online social networks; journalistic practices; intelligent metadata processing

## 1. Introduction

The increasing digitalization of our world offers significant opportunities for groundbreaking investigative journalism, new models of cross-border collaborative reporting, and access to treasure troves of knowledge and diverse sources at a mouse-click [1]. However, journalists struggle every day to cope with the overwhelming amount of information that emerges online. This combined with a time pressure to verify information as quickly as possible has caused a need for tools that can provide automatic or semi-automatic assistance to arise. Social Network Analysis (SNA) is a field that researchers turn to in order to build tools that can assist journalists in investigating topics disseminated through social media platforms by observing the propagation of claims and rumors, the discussions around the claims and rumors, and interactions between users. Coupled with intelligent methods that extract and process metadata, these tools can provide disinformation-related cues to journalists and fact-checkers and become vital in the daily activities of these professionals.

Networks are complex systems of actors, referred to as nodes, interconnected via relationships called edges. On social media, a node can be an account (i.e., a user, page, or group), a URL (i.e., an article or media item), or a keyword (i.e., a hashtag). When two nodes interact (i.e., when a Twitter account retweets a tweet of another Twitter account), an edge is formed between them. The usefulness of network visualizations is to investigate trends

and events as a whole. The challenging part of analyzing a social network is identifying the nodes and relationships that are worth investigating further.

An essential feature that makes social network analysis important for combating disinformation is that false news spreads faster than real news through online platforms involving many users and creating large networks [2]. For example, a CBC journalist [3] posted a wrong claim that identified the attacker of an attack in Toronto in 2018 as "angry" and "Middle Eastern" at the same time as another journalist who posted a claim correctly identifying the attacker as "white". It turned out that the misleading tweet identifying the attacker as Middle Eastern received far more engagement than the accurate one roughly five hours after the attack. A network emerged rapidly around the false claim, and users were quick to disseminate it. The visualization of a network involving many accounts and their interactions may reveal those accounts that try to influence the public with certain views.

During critical events, such as the 2016 US presidential election and the outbreak of the COVID-19 pandemic, fabricated information was disseminated through social media to deceive the public. Several works revealed the roles of bots (i.e., automated accounts posing as real users) in the spread of misinformation [2,4]. Their characteristics were excessive posting via the retweeting of emerging news and tagging or mentioning influential accounts in the hope they would spread the content to their thousands of followers [5]. A need to detect inauthentic users led to investigating the posting activities, interactions, and spreading behaviors. Network analysis and visualization techniques could be valuable for detecting such inauthentic accounts based on their behaviors in a network that differentiate them from those of real users.

In this work, we present the MeVer NetworkX analysis and visualization tool. The tool's development was motivated by a need to follow complex social media conversations and to gain insights about how information is spreading in networks and how groups frequently communicate with each other and form communities. The tool falls in the scope of assisting journalistic practices and, more precisely, helping journalists retrieve specific and detailed information or form a comprehensive view around a complex online event or topic of discussion. The tool aggregates publicly available information on accounts and disseminated messages and presents them in a convenient semantically enriched network view that is easy to navigate and filter, aiming to overcome the critical challenge of unstructured data on the Web. We focused on implementing a clear and straightforward navigation with no overlap among communities to provide users with easy-to-digest visualizations. A multitude of implemented features provide insights into the propagation flows and the behaviors of accounts. The primary functionality of the tool is to highlight suspicious accounts worth investigation, which could potentially speed up manual analysis processes. Finally, the tool is among the few to support three social media platforms (Twitter, Facebook, and Telegram), and its modular nature makes it extensible to more platforms. With this work, we aim to leverage intelligent metadata extractions, processing, and network science to endow journalists and fact-checkers with advanced tools in their fight against disinformation.

## 2. Related Work

A significant challenge in creating useful social graphs for the analysis of online phenomena relates to the process of data collection. The challenge is that users need to have specialized knowledge to collect data and that platforms have limitations on available data. Another aspect that strictly relates to social graph analysis is the field of bot/spammer detection. A common tactic for spreading disinformation quickly and widely is to create fake accounts that pretend to be authentic. Research in this field revealed that these accounts have certain characteristics and behaviors that lead to their automatic detection. In the following sections, we list the visualization tools introduced in the literature with a brief description of their functionalities and limitations.

### 2.1. Data Collection and Analysis

A prerequisite for creating social graphs is the collection of actors (nodes) and relationships (edges) for a query topic. In online social networks, such as Facebook and Twitter, actors can be accounts, links, hashtags, and others and edges can represent connections.

The Digital Methods Initiative Twitter Capture and Analysis Toolset (DMI-TCAT) [6] is a toolset for capturing and analyzing Twitter data. It relies on the Twitter search API to download tweets (from the last 7 days due to Twitter's rate limits) based on a search term. It provides some basic statistics on a collected dataset (the number of tweets with URLs, hashtags, and mentions; the number of tweets/retweets; and the numbers of unique users in the dataset). It creates different networks (users, co-hashtags, users–hashtags, and hashtags–URLs) and supports exporting them to the GEXF (Graph-Exchange XML Format) for visualizations. Similarly, for Twitter data, a component called Twitter SNA [7] was developed as part of the InVID-WeVerify verification plugin [8], which supports the collection of Twitter data. A plugin component transforms collected data into a format that is suitable for network visualizations and supports exports to the GEXF. CrowdTangle (https://www.crowdtangle.com/ accessed on 8 April 2022) is a tool that supports data collection for building Facebook graphs. It provides a user with an export functionality with which posts by public Facebook pages and groups are listed and accompanied by metadata. While most tools are built focusing on the collection of data from a specific platform, the open-source 4CAT Capture and Analysis Toolkit [9] (4CAT) can capture data from a variety of online sources, including Twitter, Telegram, Reddit, 4chan, 8kun, BitChute, Douban, and Parler.

An advantage of the presented tool is that it is already integrated with the Twitter SNA component (which is currently accessible through authentication and is reserved for fact checkers, journalists, and researchers to avoid misuse) of the InVID-WeVerify verification plugin so that users can automatically trigger 4CAT with query campaigns they want to investigate.

### 2.2. Bot/Spammer Detection

Significant research has been conducted to identify the spread of disinformation and spam on OSNs, especially on Twitter. Recent work proposed features based on an account's profile information and posting behaviors and applied machine-learning techniques to detect suspicious accounts. The authors in [10] examined tweet content itself and included information about an account that posted a tweet as well as n grams and sentiment features in order to detect tweets carrying disinformation. Similarly, in [11], the authors attempted to find the minimum best set of features to detect all types of spammers. In [12], a hybrid technique was proposed that uses content- and graph-based features for the identification of spammers on the platform Twitter. In [13], the authors proposed various account-, content-, graph-, time-, and automation-based features, and they assessed the robustness of these features. Other similar machine-learning techniques were proposed in [14–16]. In [17], the authors focused on the detection of not just spam accounts but also on regular accounts that spread disinformation in a coordinated way. In [18], a different methodology was followed based on a bipartite user–content graph. This work assumed that complicit spammers need to share the same content for better coverage. Shared content is also a more significant complicity signal than an unsolicited link on Twitter. The user similarity graph consisted of nodes as users and edges that represented the similarity between the users. Finally, a complete survey of recent developments in Twitter spam detection was presented in [19]. A proposed tool provided users with a convenient mechanism for inspecting suspicious accounts, leveraging features introduced in the literature. However, the automatic algorithms for detecting spam accounts are not yet part of the tool.

### 2.3. Visualization Tools

One of the most popular and most used open-source software options for network visualization and analysis is Gephi (https://gephi.org/ accessed on 8 April 2022). It

provides a multitude of functionalities for the easy creation of social data connectors to map community organizations and small-world networks. Gephi consists of many functionalities that provide users the ability to visualize very large networks (up to 100,000 nodes and 1,000,000 edges) and to manipulate the networks using dynamic filtering and SNA methods. Other network visualization tools include GraphVis (https://networkrepository.com/graphvis.php accessed on 8 April 2022), which is for interactive visual graph mining and relational learning, and webweb, (https://webwebpage.github.io/ accessed on 8 April 2022) which is for creating, displaying, and sharing interactive network visualizations on the web. ORA is a toolkit for dynamic network analyses and visualizations that supports highly dimensional network data. It is a multi-platform network toolkit that supports multiple types of analyses (e.g., social network analyses using standard social network metrics; examinations of geo-temporal networks; identifications of key actors, key topics, and hot spots of activity; and identifications of communities). NodeXL is an extensible toolkit for network overviews, discoveries, and exploration and is implemented as an add-on to the spreadsheet software Microsoft Excel 2007. It supports both the data import process and analysis functionalities, such as the computation of network statistics and the refinement of network visualizations through sorting, filtering, and clustering functions [20].

A recently introduced open-source interface for scientists to explore Twitter data through interactive network visualizations is the Twitter Explorer [21]. It makes use of the Twitter search API with all the limitations (number of requests per 15 min and tweets from the last seven days) to collect tweets based on a search term and analyze them. It includes a Twitter timeline of the collected tweets, creates interaction networks and hashtag co-occurrence networks, and provides further visualization options. A tool that visualizes the spread of information on Twitter is Hoaxy (https://hoaxy.osome.iu.edu/ accessed on 8 April 2022). This lets users track online articles posted on Twitter, but only those posted within the last seven days. A user can add a search term and visualize the interactions of at most 1000 accounts that share the term. This tool creates a graph in which each node is a Twitter account and two nodes are connected if a link to a story passes between those two accounts via retweets, replies, quotes, or mentions. Hoaxy uses the Botometer score for coloring the nodes, which calculates the level of automation an account presents using a machine-learning algorithm trained to classify.

Looking into more specialized tools, Karmakharm et al. [22] presented a tool for rumor detection that can continuously learn from journalists' feedback on given social media posts through a web-based interface. The feedback allows the system to improve an underlying state-of-the-art neural-network-based rumor classification model. The Social Media Analysis Toolkit [23] (SMAT) emerged from the challenge of dealing with the volume and complexity of analyzing social media across multiple platforms, especially for researchers without computer science backgrounds. It provides a back end data store that supports different aggregations and supports exporting results to easy user-friendly interfaces for fast large-scale exploratory analyses that can be deployed on a cloud. Significant research has focused on misinformation on the Twitter platform. BotSlayer [24] is a tool that detects and tracks the potential amplification of information by bots on Twitter that are likely coordinated in real time. Reuters Tracer [25] is a system that helps sift through noise to detect news events and assess their veracities. Birdspotter [26] aims to assist non-data science experts in analyzing and labeling Twitter users by presenting an exploratory visualizer based on a variety of computed metrics.

Our proposed tool provides several functionalities that are similar or complementary to the existing tools. Later in the paper, we present a comparison of our tool with Gephi and Hoaxy.

## 3. MeVer NetworkX Tool

The proposed tool is a web-based application for the visualization of Twitter, Facebook, and Telegram graphs. Each user submits an input file and provides their email, and a link

to a resulting graph is sent to them as soon as the processing of that file has been completed. This asynchronous means of returning the results is considered more acceptable by users, especially in cases of large graphs for which the processing time is long (several minutes). Figure 1 illustrates an overview of the tool, which is mainly separated into a server-side analysis and user interface.

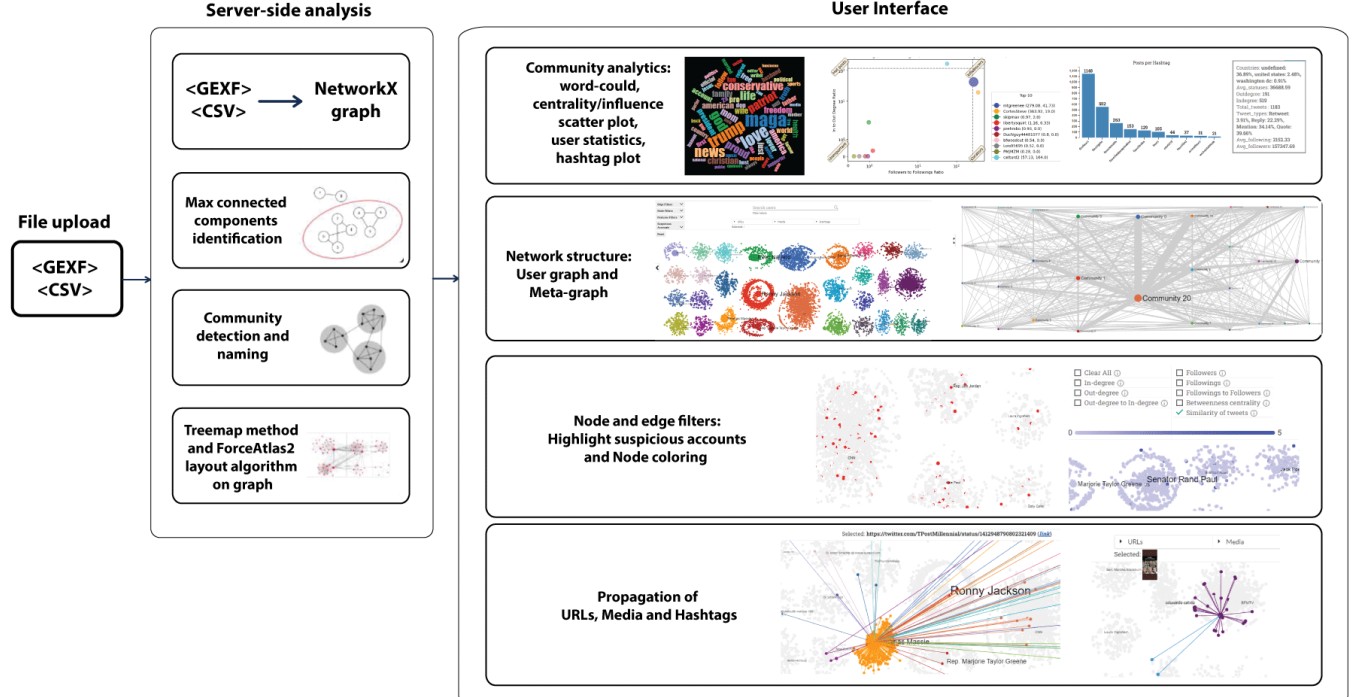

**Figure 1.** Overview of the MeVer NetworkX tool.

### 3.1. Input Files

Twitter and Telegram analyses involve GEXF files. This well-known format started in 2007 within the Gephi project to describe complex network structures and their associated data and dynamics. To build Facebook graphs, a user needs access to CrowdTangle and then he/she can export the data in CSV files.

**Twitter.** The input GEXF file of a Twitter graph contains the nodes and edges of the graph along with their attributes. The required field for a node is an ID, and the edges need the source of the edge (i.e., the ID of the node from which the edge starts and the "target" of the edge (i.e., the node's id to which the edge points)). Additional attributes are used to build plots, statistics, filters, and other functionalities. The graph contains three types of nodes visualized with different shapes: users visualized as circles, URLs visualized as stars, and hashtags visualized as rhombuses. The tool supports four edge types: retweets, quotes, mentions, and replies. Table 1 presents a list of required and optional fields. Account-related attributes (e.g., screen names and numbers of accounts following) are associated with nodes, while tweet-related attributes (e.g., retweets, texts, and hashtags) are associated with edges and characterize the interactions between nodes.

**Table 1.** Twitter's attributes for building the graph and features. All fields marked with an asterisk (*) are required.

| Attribute | Description | Type |
|---|---|---|
| Node ID * | Unique ID | Node |
| Type of node | Set to true if the node is a user node | Node |
| Tweet ID | Unique tweet ID | Node |
| Screen name | Screen name, handle, or alias that a user identifies themselves as; screen_names are unique but subject to change. | Node |
| Created at | UTC time when this tweet was created. Example: | Node |
| User description | User-defined UTF-8 string describing their account. | Node |
| Names | Name of the user as they have defined it in Twitter. | Node |
| Number of followers | Number of followers this account currently has | Node |
| Location | User-defined location for this account's profile. | Node |
| Number of accounts following | Number of users this account is following | Node |
| Verified | When true, indicates that the user has a verified account. | Node |
| Number of statuses | Number of tweets (including retweets) issued by the user. | Node |
| Profile image | HTTPS-based URL pointing to the user's profile image. | Node |
| Background image | HTTPS-based URL pointing to the standard Web representation of the user's uploaded profile banner. | Node |
| Edge ID | Unique ID | Edge |
| Source * | Node that the edge starts at | Edge |
| Target * | Node that the edge points to | Edge |
| Tweet ID | Unique tweet ID | Edge |
| Retweet | Whether the edge is a retweet | Edge |
| Reply | Whether the edge is a reply | Edge |
| Mention | Whether the edge is a mention | Edge |
| Quote | Whether the edge is a quote | Edge |
| Created at | UTC time when this tweet was created. | Edge |
| Number of retweets | Number of times this tweet has been retweeted. | Edge |
| Number of favorites | Approximately how many times this tweet has been liked by Twitter users. | Edge |
| Text | Actual UTF-8 text of the status update. | Edge |
| Hashtags | Hashtags that have been parsed out of the tweet text. | Edge |
| URLs | URLs included in the text of a tweet. | Edge |
| Media | Media elements uploaded with the tweet. | Edge |

**Facebook.** CrowdTangle tracks only publicly available posts and extracts data in CSV files. The CSV files contain one public Facebook post per line with metadata about the page/group that posted it and the post itself. We considered two types of node: groups and resources (URLs, photos, or videos); the interactions among them are defined as the edges of a graph. The nodes are visualized with different shapes, namely a circle for a Facebook group/page, a star for an article, a rhombus for a photo, and a square for a video. An edge is created from a group/page node made into a resource node when the group/page shares a post containing the resource node's link. When multiple groups share resources, multiple edges are created for the resource node. Metadata that refers to Facebook pages/groups are used as node attributes, while information related to Facebook posts that contain resources (URLs, photos, or videos) is associated with edges since a resource might be associated with multiple pages/groups. Table 2 summarizes and explains the attributes used for building Facebook graphs.

**Telegram.** The Telegram graph has three node types visualized, each with a different shape: (i) users (circles), (ii) URLs (stars), and (iii) hashtags (rhombuses). Edges represent occurrences of hashtags and URLs within the text field of a message sent by a user through a channel. Based on how the Telegram ecosystem is conceived and constructed, it's not possible to determine the actual Telegram account that used the channel as a mean to communicate with its subscribers. However, channel administrators can use the author's signature feature in order to include the first and last name in the signature of each message they send. Such an indicator cannot lead anyone to a unique identification of the Telegram account that has been used to send certain messages containing a specific signature. Due to

this, the user node visualized with a circle in the graph corresponds to the channel name and not the author of the messages.

**Table 2.** Facebook's attributes for building the Facebook graph and features. All fields marked with an asterisk (*) are required.

| Attribute | Description | Type |
|---|---|---|
| Page/group name | Name of the page/group that posted | Node |
| User name | Username of the page/group | Node |
| Facebook ID * | ID of the page/group | Node |
| Likes at posting | Number of likes of the page/group at the time of posting | Node |
| Followers at posting | Number of page/group followers at the time of posting | Node |
| Type | Types of links (articles, photos, and videos) included in the post | Node |
| Resource * | Link included in the post | Node/edge |
| Total interactions | Total number of all reactions (likes, shares, etc.) | Edge |
| Message | Message written in the post | Edge |
| Created | Time the post was published | Edge |
| Likes | Number of likes on the post | Edge |
| Comments | Number of comments on the post | Edge |
| Shares | Number of shares of the post | Edge |
| Love | Number of love reactions on the post | Edge |
| Wow | Number of wow reactions on the post | Edge |
| Haha | Number of haha reactions on the post | Edge |
| Sad | Number of sad reactions on the post | Edge |
| Angry | Number of angry reactions on the post | Edge |
| Care | Number of care reactions on the post | Edge |

At this point, it is really important to underline and remark that the researchers anonymized the data before operating or storing it with a cryptographic hash function named *BLAKE2*, which is defined in RFC 7693 (https://datatracker.ietf.org/doc/html/rfc7693.html accessed on 8 April 2022). If a message contains one or more hashtags and/or one or more URLs, a node is created for each entity that can be extracted, and an edge connecting each couple of nodes is created too. As per our ethical considerations, only open-source and publicly available information was gathered and analyzed. This ethical deliberation should not be interpreted as an actual obstacle or limit, given that disinformation actors prefer these public Telegram venues. The entities that can be extracted from each message are listed in Table 3.

**Table 3.** Telegram's attributes for building the Telegram graph and features. The field marked with an asterisk (*) is optional.

| Attribute | Description | Type |
|---|---|---|
| ID | Unique identifier of the message within the channel | Edge |
| Message link | URL to the message in the object | Edge |
| Hashtags | Hashtags included in the message | Node/edge |
| Links | Links included in the message | Node/edge |
| Timestamp | The time at which the message was sent | Edge |
| Message | Text of message | Edge |
| Author's signature * | First and last name of the author of the message | Edge |
| Views | Number of times a message was viewed | Edge |

### 3.2. Features and Functionalities

#### 3.2.1. Individual Account and Post Inspections

Users can click on individual accounts of interest and obtain information about various account statistics, the most influential nodes they are connected to, and the communities they interact with the most (Figure 2a). Additionally, a user can focus the visualization on

the interactions (edges) of an account in a graph (Figure 2c). Finally, the text of the posts (only for Twitter) made by a selected user are presented (Figure 2b).

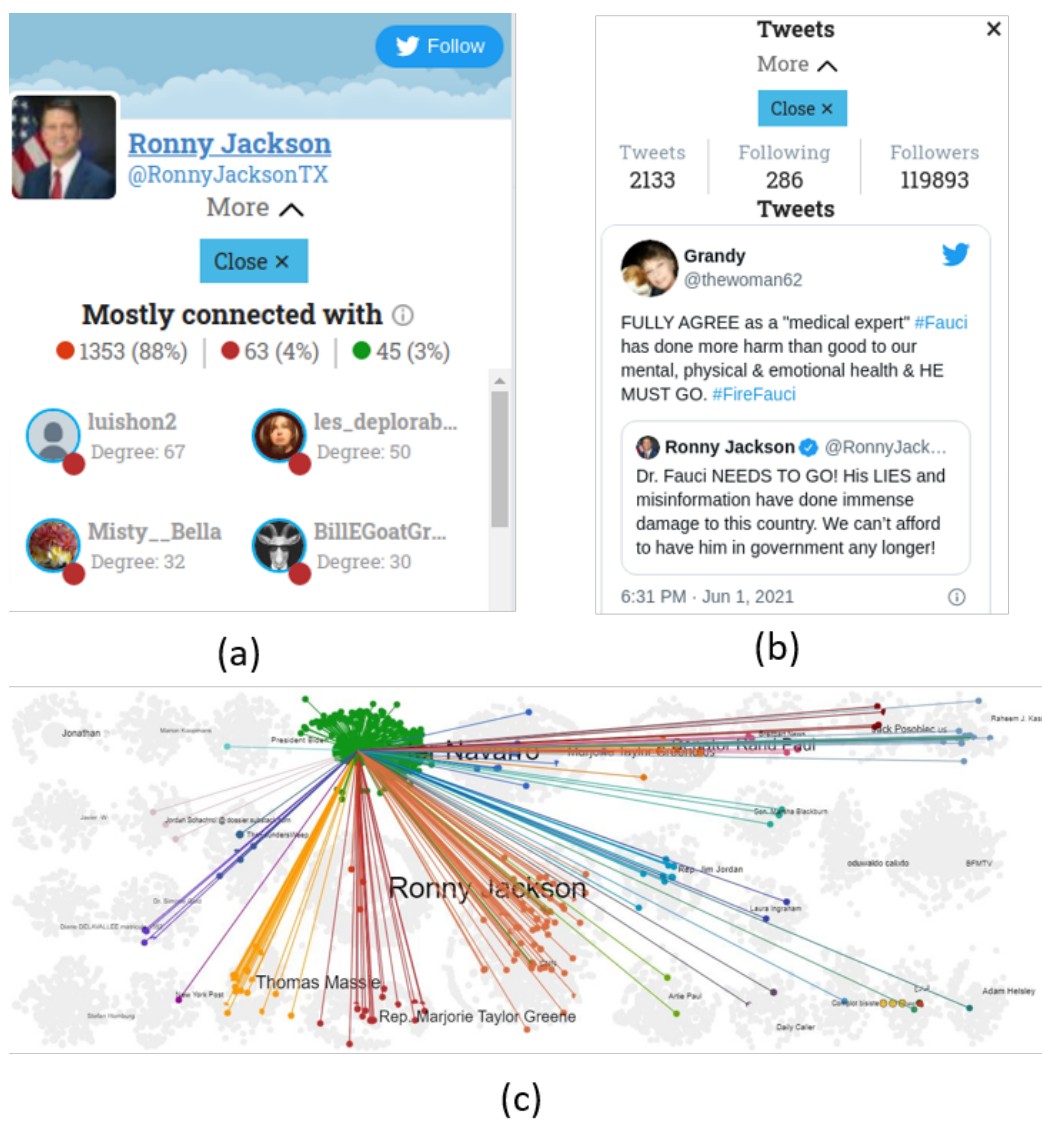

**Figure 2.** Individual user inspection. (**a**) Information about the user, (**b**) text of the tweets made by the user and (**c**) the interactions of this user.

### 3.2.2. Community Detection

We identified all the connected components and then discarded those with fewer than three nodes. We undertook this mainly because in the vast majority of cases, communities with very few nodes do not provide helpful insights on a network. In such situations, the Louvain [27] algorithm was applied to perform community detection on a graph. The formed communities were named using the names of the top three nodes based on their degrees (number of edges a node has). Nodes with high degrees can provide insights about the rest of a community as they are usually centered around a specific topic. The last step before the visualization refers to the positioning of nodes within communities and then positioning the communities in the graph. To this end, we followed a two-step procedure that is an adaptation of the TreeMap methodology [28] in combination with the ForceAtlas2 [29] layout algorithm, which prevents overlapping nodes and provides a clear and readable graph. The main idea of the TreeMap positioning method was the division of the screen area into rectangles of different sizes and the assignment of each

community into a rectangle taking into account the number of nodes that belonged to the corresponding community. To adapt this method to our needs, we implemented Algorithm 1 to order communities in order to position the largest community at the center of the screen and the rest of the communities around it, taking into account the number of interactions and avoiding overlapping. Next, Algorithm 2 allocated the screen area to each community so that larger communities spanned larger areas. Algorithm 3 performed the above procedures and executed a Python implementation of the *TreeMap* algorithm to position the communities and then executed a Python implementation of the ForceAtlas2 (https://github.com/bhargavchippada/forceatlas2 accessed on 8 April 2022) layout algorithm to position the nodes within the communities.

---

**Algorithm 1** Order communities. Largest community at the center of the graph. Communities with most interactions are closest.

---

1: **procedure** COMMUNITY_LIST($coms, edges$)
2:     $k \leftarrow 0$
3:     $max\_com\_id \leftarrow get\_largest\_com(coms)$
4:     $middle \leftarrow len(coms)/2$
5:     $order\_coms[middle] \leftarrow coms[max\_com\_id]$
6:     $temp\_id \leftarrow max\_com\_id$
7:     **while** $len(coms) > 0$ **do**
8:         **for** i in edges **do**
9:             $max\_edges\_id \leftarrow find\_max\_edges(i, temp\_id)$
10:         **end for**
11:         $k \leftarrow k + 1$
12:         **if** add to left **then**
13:             $order\_coms[middle - k] \leftarrow coms[max\_edges\_id]$
14:         **else**
15:             $order\_coms[middle + k] \leftarrow coms[max\_edges\_id]$
16:         **end if**
17:         $temp\_id \leftarrow max\_edges\_id$
18:         $remove(coms[max\_edges\_id])$
19:     **end while return** $order\_coms$
20: **end procedure**

---

**Algorithm 2** Allocation of areas to communities based on the communities' sizes.

---

1: **procedure** ALLOCATE_AREA($order\_coms\_sizes, width, height$)
2:     $minimum\_size \leftarrow round(width * height/len(order\_coms\_sizes)/2)$
3:     $total\_area \leftarrow width * height - minimum\_size * len(order\_coms\_sizes)$
4:     **for** size in order_coms_sizes **do**
5:         $com\_sizes \leftarrow (size * total\_area/total\_size) + minimum\_size$
6:     **end for return** $com\_sizes$
7: **end procedure**

---

**Algorithm 3** Positioning of communities and nodes.

---

1: **procedure** POSITIONING($coms, edge$)
2:     $width \leftarrow Screen\_width$
3:     $height \leftarrow Screen\_height$
4:     $order\_coms \leftarrow community\_list(coms, edge)$
5:     $com\_sizes \leftarrow allocate\_area(order\_coms\_sizes, width, height)$
6:     $rectangles \leftarrow treemap(order\_coms, com\_sizes)$
7:     $x\_node, y\_node \leftarrow ForceAtlas2(rectangles)$ **return** $x\_node, y\_node$
8: **end procedure**

Figure 3 shows an example graph. A user can obtain insights about the number of nodes and edges in a graph and easily identify the most influential nodes since they are the largest ones and their names are shown.

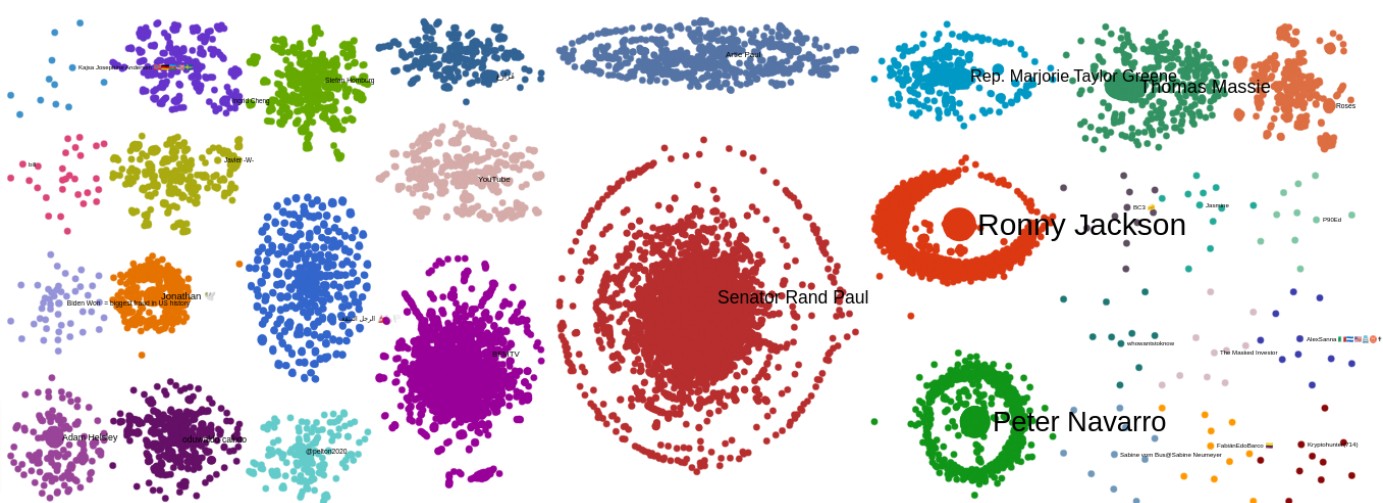

**Figure 3.** Visualization of example graph.

### 3.2.3. Community Analytics

Analytics on the detected communities help users gain better intuitions about them. Statistics per community provide summaries of each community's users. Word clouds and hashtag plots present the most frequently used hashtags and help users identify the most prominent topics per community. The frequencies of the most popular hashtags shared within a community are plotted. With regards to the posting activities of users, a time series plot shows the number of tweets shared per day by users of the community, revealing activity patterns.

Finally, a centrality/influence scatter plot is produced for the top 10 users of each community. The x axis shows the number of followers with respect to the number of followings, and the y axis corresponds to the ratio of incoming to outgoing interactions. Betweenness centrality is illustrated with bubbles; the larger the bubble, the higher the value of the feature calculated for a node. This plot helps identify community users that have essential roles in the spread of information (a high value of betweenness centrality) in correlation with their popularity rate (the x axis) and interaction rate (the y axis). The accounts in this plot are divided into four categories based on their positions. (i) *hot posts*: These have equal or smaller number of followers than followings and can be considered "regular" users (not popular). Their tweets have a strong influence on the spread of information as they have attracted the interest of other users. (ii) *Influencers*: These have higher numbers of followers than followings and can be considered popular. Their tweets have attracted the interest of other users, and their posts play vital roles in a community's topic and the spread of information. (iii) *Curators*: These have higher numbers of followers than followings and are regarded as popular. They have high posting activity levels as they usually post tweets and reference other accounts more than the opposite. Their beliefs are essential parts of a community's topic. (iv) *Unimportant*: These accounts have an equal or smaller number of followers than followings and are not popular. Their tweets do not attract other users' interest. Figure 4 presents an example of the analytics for a Twitter community. In the case of FB graphs, a heatmap of reactions per community shows the distribution of interactions on the posts of the top 10 Facebook pages/groups of the community based on the average number of total interactions per post.

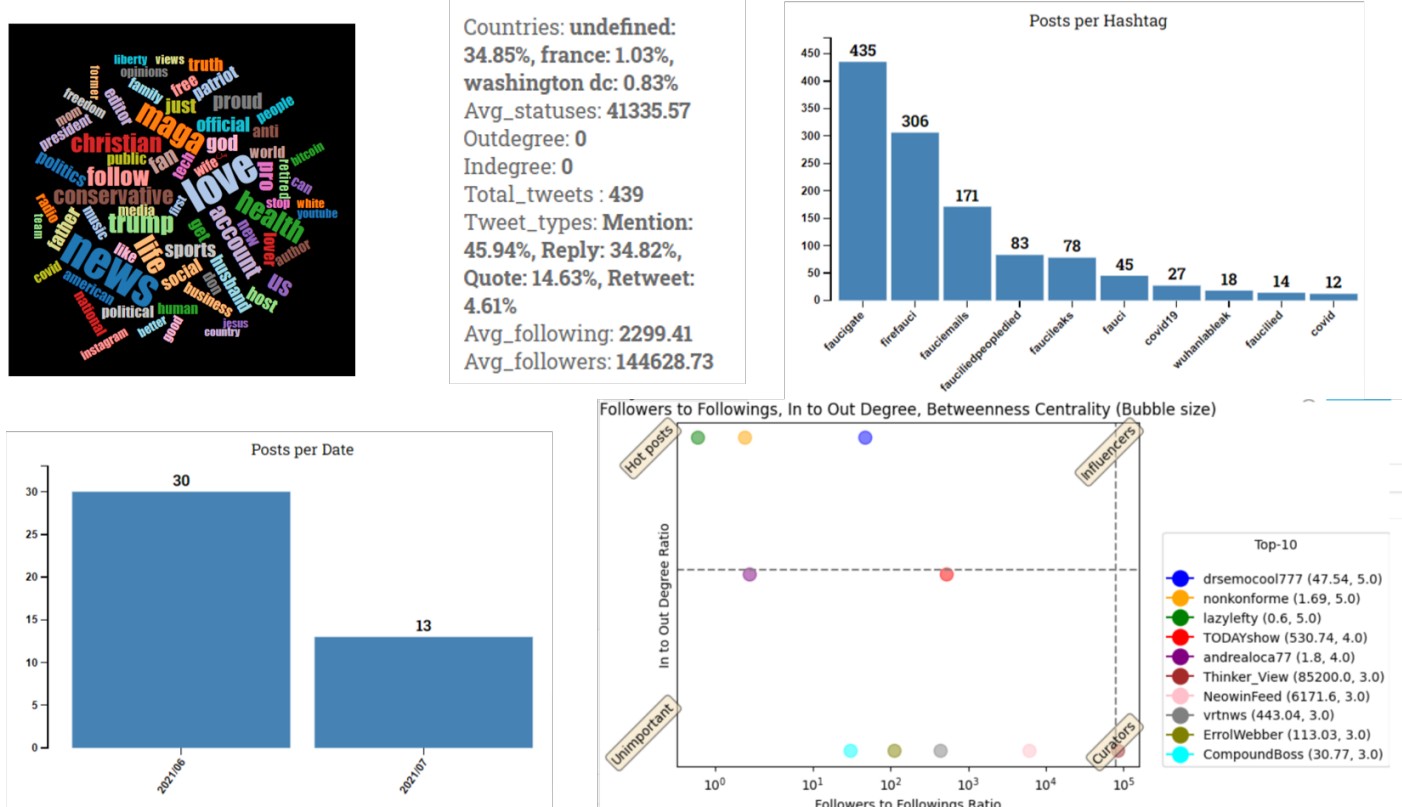

**Figure 4.** Community analytics: word cloud, user statistics, hashtag plot, tweet-posting plot, and centrality/influence scatter plot.

### 3.2.4. Propagation of URLs, Media, and Hashtags

To help study the spread of content in a network, we collected the top 10 URLs, media items, and hashtags in a network, and for each of them, we presented the interactions between the users. It is worth noting that there were items that appeared only inside a community and others that were disseminated across different communities.

### 3.2.5. Metagraph

The metagraph tool provides a higher-level representation of extracted communities and their interactions using a metagraph view: nodes correspond to communities, and their edges are weighted based on the interactions among the accounts of the respective communities. Wider edges between two communities mean a higher number of interactions across them. This view provides rich information about the relationships between the accounts of two and more communities. The metagraph view aims to reveal the degree to which the topics of different communities are related to each other. Wider edges indicate a higher correlation of topics.

### 3.2.6. Node and Edge Filters

Filters aim to offer a more targeted investigation and allow a user to limit information based on specific criteria. We grouped filters according to their types and placed them at the top left of the graph for easy navigation. Users with few interactions (a low degree) might not be of interest during an investigation and may therefore be removed using the min log-scaled degree filter. Additionally, if a user needs to investigate the interactions between accounts during a specific time interval, they can use the interactions date filter. Beyond node-based filters, there are also three types of edge filters depending on the type of edge: tweet edges, URL edges, and hashtag edges. For example, a tweet edge filter maintains (or removes) edges that are retweets, mentions, replies, and quotes. Users may also combine filters, which offers significant versatility during investigations.

### 3.2.7. Node Coloring

The node coloring tool exposes eight features characterizing an account's (node's) behavior. When a user selects one such feature, node colors are printed in a color scale from white (low) to dark blue (high) based on their values on this feature. These features include the following.

- `In-degree` quantifies the popularities of nodes, i.e., how much a node is referenced/linked to by others.
- `Out-degree` shows the extroversion of nodes. A node that references many other nodes on its tweets has a high value in this feature.
- `Out-degree to in-degree` is the ratio of out- to in-degree. Accounts that regularly reference other nodes in their tweets and are rarely referenced by others have high values in this feature. Accounts that are both extroverted and popular have low values.
- `Followers` is the number of account followers.
- `Followings` is the number of accounts that the account follows.
- `Followings to followers` is the ratio of the number of followings to the number of followers. This feature quantifies how popular an account is (low value), how selective it is with its followings (low value), and how likely it is to follow back (high value).
- `Betweenness centrality` captures the role of a node in the spread of information across a network. Higher values indicate more important roles.
- `Similarity of tweets` shows how similar the posts of an account are. A node with a high value in this feature regularly posts similar content on its tweets.

### 3.2.8. Highlight Suspicious Accounts

Finally, we implemented six features that, when combined with three of the above, indicate suspicious accounts spamming or spreading disinformation. Inspired by features presented in the literature to train machine-learning models and detect spam posts [14], we implemented a set of features that support the interactive exploration of a dataset to find accounts that were worth further investigation. The features were normalized using min-max normalization. To calculate extreme values on these features and subsequently highlight accounts with such values, we used quartiles and boxplots. The accounts were highlighted in red on the graph, providing a semi-automatic identification of suspicious accounts. The implemented features are listed below.

- `Following rate` is the ratio of the number of followings to the number of days since an account was first created.
- `Status rate` is the ratio of the number of posts to the number of days since an account was created.
- `Average mentions per post` shows the average number of mentions in an account's tweets. A common strategy for spreading disinformation is mentioning many accounts in tweets.
- `Average mentions per word` shows the average number of mentions in a tweet's text. The tactic of posting tweets with many mentions and a single hashtag is often regarded as spam-like or suspicious. This feature is normalized to the total number of posts.
- `Average hashtags per word` calculates the average number of hashtags in a tweet's text.
- `Average URLs per word` calculates the average number of URLs in a tweet's text.

Figure 5 illustrates an example of a Twitter graph for a Fauci use case. At the left of the figure, the betweenness centrality node coloring is applied to highlight nodes with high influence over the flow of information in the graph and that are worth investigating. At the right of the figure, all highlights of suspicious account features are selected, and 425 accounts are highlighted as suspicious, limiting users who are worth evaluating and offering a user a clue of where to start an investigation.

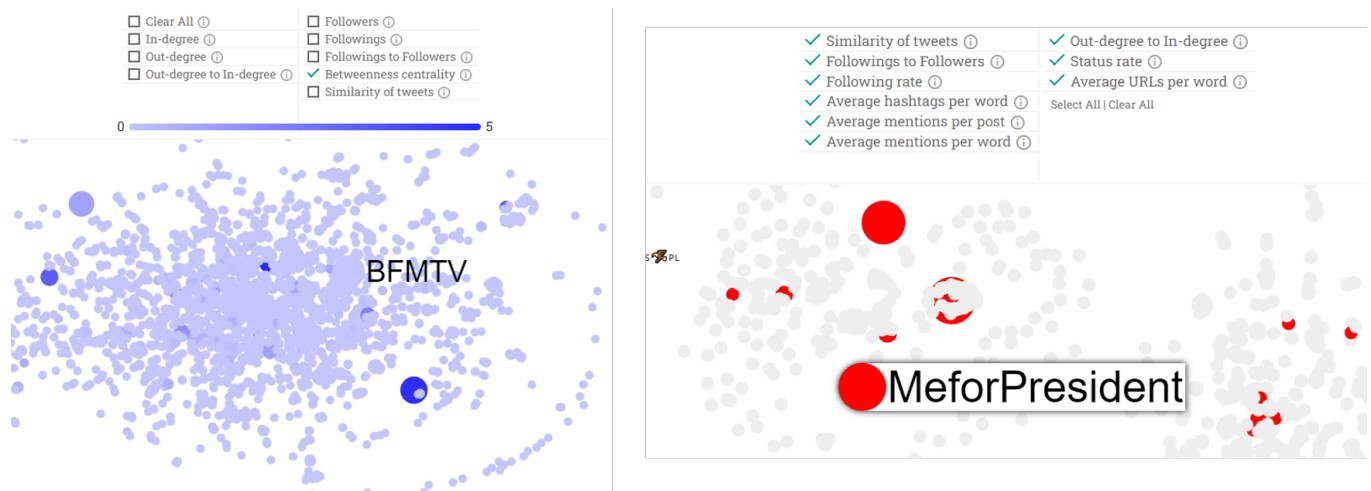

**Figure 5.** Example of the node-coloring filter (**left**) and highlighting suspicious accounts filter (**right**).

Table 4 summarizes the features/functionalities supported per platform. Although we aimed to adapt all the developed features on all three platforms, the available data restricted the implementation of features in some cases.

**Table 4.** List of developed features and the platforms that are supported.

| Features/Functionalities | Twitter | Facebook | Telegram |
|---|---|---|---|
| Communities | ✓ | ✓ | ✓ |
| Individual account inspections | ✓ | ✓ | ✓ |
| Post text | ✓ | ✗ | ✗ |
| Statistics per community | ✓ | ✓ | ✗ |
| Word clouds per community | ✓ | ✓ | ✓ |
| Centrality plots per community | ✓ | ✗ | ✗ |
| Date plots per community | ✓ | ✓ | ✓ |
| Hashtag plots per community | ✓ | ✗ | ✓ |
| Heatmaps of reactions per community | ✗ | ✓ | ✗ |
| Propagation flow of top 10 URLs | ✓ | ✓ | ✓ |
| Propagation flow of top 10 media | ✓ | ✓ | ✓ |
| Propagation flow of top 10 hashtags | ✓ | ✗ | ✓ |
| Metagraphs | ✓ | ✓ | ✓ |
| Edge Filters | ✓ | ✗ | ✗ |
| Node Filters | ✓ | ✓ | ✓ |
| Node coloring | ✓ | ✓ | ✓ |
| Suspicious accounts | ✓ | ✗ | ✗ |

## 4. COVID-19-Related Use Cases

### 4.1. Twitter Data

We collected four COVID-19-related datasets for topics for which disinformation was prominent. To collect the datasets, we used the Twitter search API, querying with the hashtags #FireFauci, #FauciGate, #Hydroxychloroquine, #BigPharma, and #GreatReset. We collected all tweets containing these hashtags posted between 1 June 2021 and 15 July 2021. Table 5 presents the dataset statistics.

**Table 5.** Statistics for the collected COVID-19-related Twitter disinformation datasets.

|  | Fauci | Hydroxychloroquine | Big Pharma | Great Reset |
|---|---|---|---|---|
| Total tweets | 18,500 | 6239 | 16,568 | 13,380 |
| Retweets | 4790 | 3597 | 9667 | 6780 |
| Quotes | 7787 | 1114 | 2281 | 2615 |
| Tweets with replies | 4696 | 1046 | 3579 | 3037 |
| Tweets with mentions | 4926 | 2439 | 5609 | 4884 |
| User-posted tweets | 11,155 | 4310 | 10,474 | 8175 |
| Total users in tweets | 18,310 | 7078 | 18,175 | 14,716 |

The Fauci and hydroxychloroquine cases came after the Washington Post and BuzzFeed News filed Freedom of Information Act requests for Dr. Anthony Fauci's emails, published that correspondence on 1 June 2021, and showed how Dr. Anthony Fauci navigated the early days of the COVID-19 pandemic. The emails contained discussions about what Fauci was told on the origins of the coronavirus, what he knew about the drug hydroxychloroquine, and what he said about the use of face masks. Apart from informing the public, this email disclosure led to the propagation of misleading facts about COVID-19 by decontextualizing parts of the discussions. Fauci's political opponents and several conspiracy theorists took the opportunity to spread their beliefs on social networks by sharing out-of-context claims.

Another conspiracy theory that gained popularity was the Big Pharma theory. A group of conspiracy theorists claimed that pharmaceutical companies operate for sinister purposes and against the public good. They claimed that the companies conceal effective treatments or even cause and worsen a wide range of diseases.

Finally, the Great Reset theory referred to a theory that the global elites have a plan to instate a communist world order by abolishing private property while using COVID-19 to solve overpopulation and enslave what remains of humanity with vaccines.

*4.2. Facebook Data*

We used CrowdTangle to collect Facebook posts on the aforementioned COVID-19-related use cases. We submitted four search queries in CrowdTangle using the hashtags #FireFauci, #FauciGate, #Hydrochloroquine, #BigPharma, and #GreatReset as search keywords. The search retrieved Facebook posts by public Facebook pages and groups (but not posts by Facebook users due to Facebook graph API limitations). We ended up with two datasets, Fauci and hydroxychloroquine. The Big Pharma and Great Reset topics were discarded due to very low numbers of retrieved posts. Table 6 lists the statistics for the Fauci and hydro datasets.

**Table 6.** Statistics for the collected COVID-19-related Facebook disinformation datasets.

|  | Fauci | Hydroxychloroquine |
|---|---|---|
| FB posts | 553 | 1572 |
| FB groups/pages | 352 | 984 |
| Articles | 95 | 504 |
| Photos | 109 | 264 |
| Videos | 71 | 53 |

*4.3. Telegram Data*

In order to acquire information from public Telegram channels, a specific data-acquisition system was crafted. A core component of such a system is, usually, Web Scraper, the aim of which is to parse HTML markup code and pass it through a set of selectors in order to structure the acquired information made available via a Web user interface. A relevant difference between the platform Telegram and other platforms, such as Facebook and Twitter, is the absence of an internal content-search feature. Thus, only public channels and groups can be found via

a global search feature—not the messages contained in them. In order to find public channels of interest, two different approaches were followed. The first was executing keyword-based searches via a Google custom search engine specifically designed for Telegram content (https://cse.google.com/cse?cx=004805129374225513871:p8lhfo0g3hg accessed on 8 April 2022). The second approach was running hashtag-based searches on Twitter and applying a filter to them in order to receive only results containing at least one URL referencing Telegram content. We chose to select up to three public channels per each hashtag. Specifically, we selected from the first three top tweets, (https://help.twitter.com/en/using-twitter/top-search-results-faqs accessed on 8 April 2022) which Twitter valued as most relevant (in descending order) at the time when we executed the search. For both approaches, we decided to select only the first three results provided in order both to keep the information consistent and to have a sufficiently sized dataset. Subsequently, the information populating our Telegram dataset was acquired from chats by the following identified handles: @RealMarjorieGreene, @QtimeNetwork, @cjtruth316, @trumpintel, @Canale_Veleno, @WeTheMedia, @shaanfoundation, @TerrenceK-Williams, and @zelenkoprotocol. Specifically, the first three were selected for the *#FireFauci* dataset, the second three were selected for the *#FauciGate* dataset, and the last three were selected for the *#Hydroxychroroquine* dataset. Table 7 lists the statistics for the #FireFauci, #FauciGate, and #Hydroxychloroquine datasets.

**Table 7.** Statistics for the collected COVID-19-related Telegram disinformation datasets.

|  | **FireFauci** | **FauciGate** | **Hydroxychloroquine** |
|---|---|---|---|
| Subscribers | 326,586 | 6488 | 186,236 |
| Messages | 14,762 | 181,700 | 13,422 |
| URLs | 6453 | 83,993 | 10,032 |
| Hashtags | 871 | 18,653 | 106 |

*4.4. Iterative Evaluation and Feedback*

We applied the analysis to the four collected COVID-19-related use cases, which are cases of significant value that reflect the challenge of disinformation. These use cases arose in the context of user-driven evaluation activities that took place within the Horizon 2020 WeVerify project (https://weverify.eu/ accessed on 8 April 2022). Journalists and fact-checkers participated in these evaluation activities and provided helpful feedback on the proposed MeVer NetworkX analysis and visualization tool. The users received brief guidelines on the functionalities of the tool and query files on a set of use cases collected within the project (beyond the four use cases presented here). They analyzed the query cases and provided comments/suggestions on the parts of the tool that were unclear to them and parts that would make the analysis easier to digest and more efficient. We enhanced the tool with the user feedback and came up with the final version that is presented in this paper.

**5. Analysis Using the MeVer NetworkX Analysis and Visualization Tool**

The main focus of our analysis was to simulate a scenario in which, through the tool, an end user tries to identify and inspect suspicious accounts within a given dataset graph.

*5.1. Fauci*

5.1.1. Twitter Analysis

The graph included 18,310 nodes and 27,882 edges. Different colors were assigned to the nodes of different communities. We first selected the all option of the suspicious account filter at the top of the graph. The resulting 425 accounts highlighted as suspicious were presented in the graph in red, as shown in Figure 6. We queried the Twitter search API three months after the dataset was collected, and 78 of the 425 likely suspicious accounts did not exist on Twitter anymore due to violating Twitter policies. This indicates that the tool likely correctly highlighted at least 78 accounts. Note that the fact that the remaining

347 accounts were still active 3 months after the dataset collection does not mean that they were not suspicious.

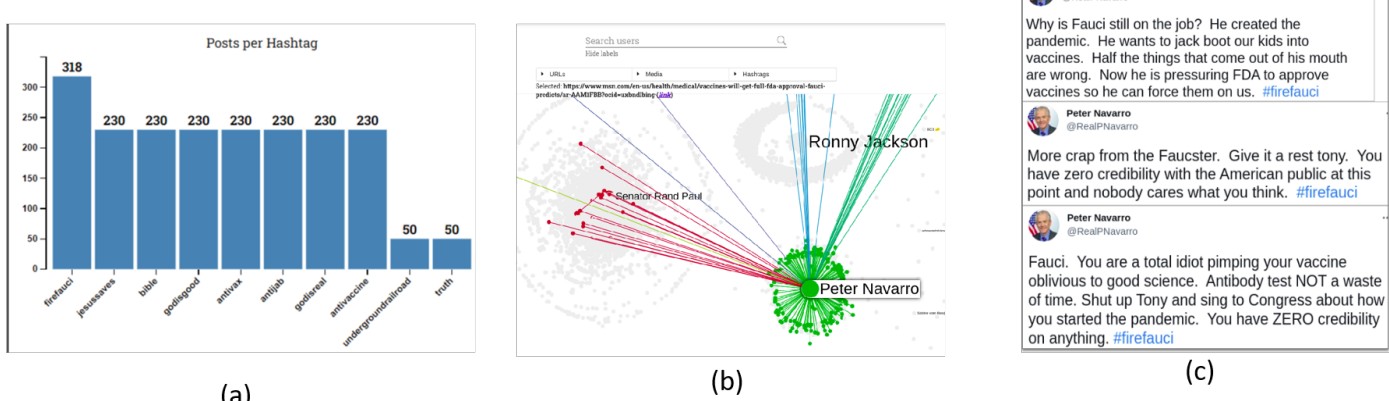

**Figure 6.** Example of a suspicious account in the Fauci analysis.

For instance, an account that was flagged as suspicious by the tool was the account with the highest number of interactions, (@*WavesofGo*), in Community 14 (Figure 6). The account had 228 interactions and was the largest circle in the community. The interactions corresponded to mentions of 228 individual accounts in the tweets, while there were no interactions toward this account from other users. The mentions referred to popular accounts, such as President Biden's @*POTUS* official account and Marjorie Taylor Greene's @*mtgreenee* account. Additionally, the similarity of tweets feature indicated that the user posted the exact tweet text each time by referencing different accounts. Table 8 shows two examples of tweets shared by this account through the network, indicating that the account was a strong supporter of Christianity and an opponent of vaccination. Twitter suspended the account due to not complying with Twitter's policies. We then inspected the statistics on Community 14. The hashtag plot revealed the community's main topics were (#JesusSaves and #AntiVax), as shown in Figure 7a.

**Figure 7.** Results of Fauci use-case analysis: (**a**) hashtag plots of Community 14, (**b**) an example of MSN article spread, and (**c**) Peter Navarro's tweets against Fauci.

**Table 8.** Tweets of *@WavesofGo* account that indicate the account was a strong supporter of Christianity and was against vaccination.

| | |
|---|---|
| "Jesus heals the sick when men had leprosy, the worst diseases back then and people were afraid to be around them. Don't take Covid Vaccine because JESUS IS A HEALER. Repent and confess to Jesus. #AntiVax #AntiVaccine #AntiJab #GodIsReal #FireFauci #Bible #GodisGood #JesusSaves" | "Praise God! YES Jesus heals the sick and did so when men had leprosy and people were afraid to be around them. That's why Christians shouldn't take Covid Vaccine because our GOD IS A HEALER. #AntiVax #AntiVaccine #AntiJab #GodIsReal #FireFauci #Bible #GodisGood #JesusSaves #God" |

Similarly, the most active account of Community 10, *Adam Helsley (@AdamHelsley1)*, was flagged as suspicious. The account only replied to other accounts with the #FireFauci hashtag, trying to campaign against Fauci. The account remained active on Twitter three months after the data collection since its behavior complied with Twitter's policies, even though its posting activity resembled that of a spammer. Our tool highlighted this as suspicious behavior. The end user is responsible for investigating further and deciding whether this account tries to adversely affect other users. This account triggered five of the nine suspicious features, namely the ratio of out-degree to in-degree, the average number of mentions per post and word, the average number of hashtags per word, and the similarity of tweet text features.

For a further investigation, we used the Botometer, a tool that checks the activity of a Twitter account and gives it a score on how likely the account is a bot. The Botometer's values range between zero and five, and the higher the value, the more likely the account is a bot. Despite the above signs, the Botometer did not classify the account as a bot.

Next, we investigated the propagation flow of the top 10 URLs, hashtags, and media items. By selecting an item, a user can monitor the item's dissemination across a network. In this case, one of the most popular URLs in the network was an *msn.com* article (https://www.msn.com/en-us/health/medical/vaccines-will-get-full-fda-approval-fauci-predicts/ar-AAM1FBB?ocid=uxbndlbing accessed on 8 April 2022), which says that Dr. Fauci advises vaccinated Americans to wear masks in areas with low COVID-19 immunization rates. This topic attracted Twitter users' interest and indicated that it was worth a further investigation. This article was mainly spread in Community 3 (Figure 7b). The account of this community interacting the most frequently was *Peter Navarro (@RealPNavarro)*. Peter Kent Navarro is an American economist and author who served in the Trump administration as assistant to the president, director of trade and manufacturing policy, and policy coordinator of the National Defense Production Act. This account has more than 161 thousands followers. He posted aggressive tweets against Fauci that spawned heated discussions on Twitter. From his 1373 total interactions, only five referenced other accounts, while in the remaining 1368, he was referenced by other accounts. Figure 7c shows his top three tweets that gained the most interactions in the network and reveal his negative attitude toward Fauci.

Finally, the word cloud plot of Community 3 provided insights into the topics of the accounts that made it up. In Figure 8a, we observe that words such as "Trump" and "patriot" frequently appeared, concluding that many accounts in this community are likely Trump supporters and explaining the reason for the attack against Fauci. Moreover, the centrality plot, shown in Figure 8b, labels Peter Navarro as an influencer.

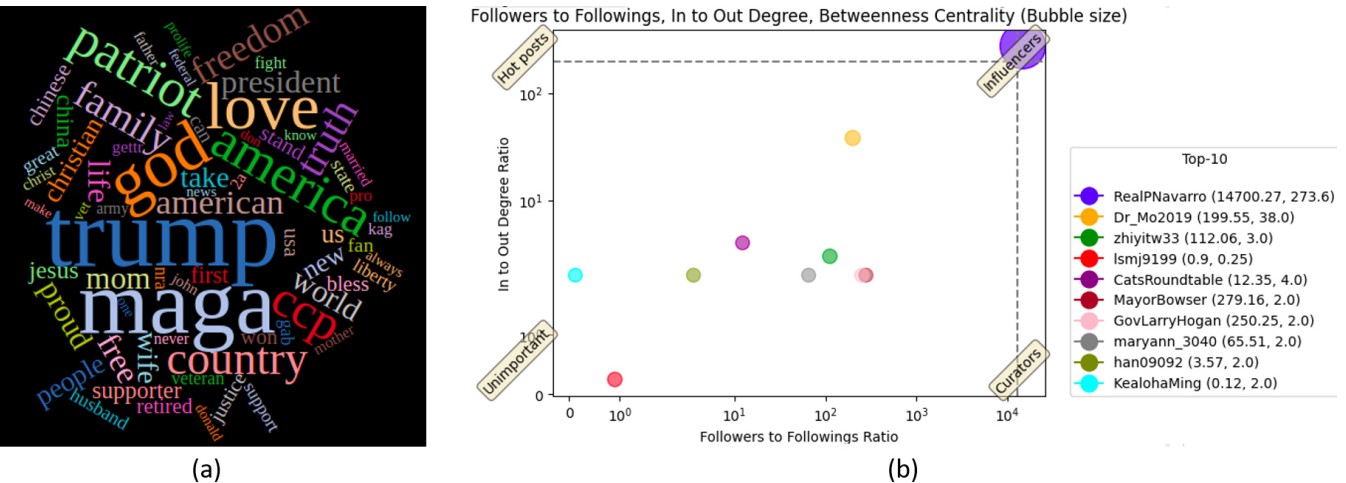

(a)                                                                  (b)

**Figure 8.** Results of Fauci use-case analysis: (**a**) word cloud of Community 3 and (**b**) centrality plot of Community 3 indicating that Peter Navarro's account stands as an influencer in the community.

5.1.2. Facebook Analysis

The 627 nodes and 553 edges derived from the Facebook Fauci dataset formed 29 communities. In all, 19 of the 29 communities consisted of three nodes, which we discarded as unimportant; we focused on the remaining 10 communities but noticed that the information diffusion among these communities was limited. Figure 9 illustrates the corresponding metagraph. We noticed that the largest community (with the ID 0) had no interactions with other communities at all.

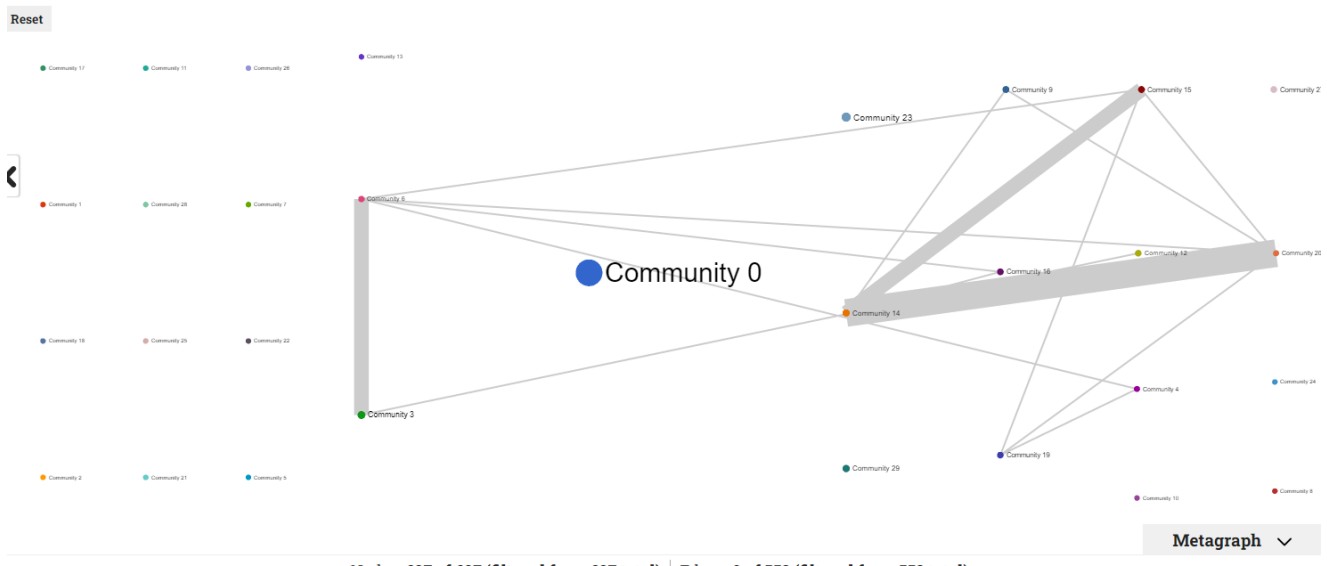

**Figure 9.** Metagraph showing that the information diffusion in the network was limited.

The node most frequently interacted with in Community 22 was an article from the Convention of States Action, which was the most-shared link in the network. The article references a poll about the aspect of parents with respect to the vaccination of their children. It states that Dr. Fauci lied to people, which led to hesitation among parents about their children's vaccination. The article was shared among the 52 nodes of Community 22, creating a network of supporters.

Using the tool, we managed to draw some interesting observations that provided insights about the network users' attitudes toward the topic. The positive reactions filter highlighted nodes that received love, wow, haha, and care reactions. Figure 10 shows the

highlighted nodes of Community 8 that were mainly around Marjorie Taylor Greene, a Republican congresswoman who sparked the #FireFauci hashtag on Twitter. The network revealed positive emotions around Marjorie Taylor Greene, which was the opposite of the aggressive attitudes toward Fauci. Additionally, through the propagation flow feature, a link of the top 10 shared links in the network pointed to a Facebook image post containing a screenshot of a Marjorie Taylor Greene post urging for Fauci's dismissal.

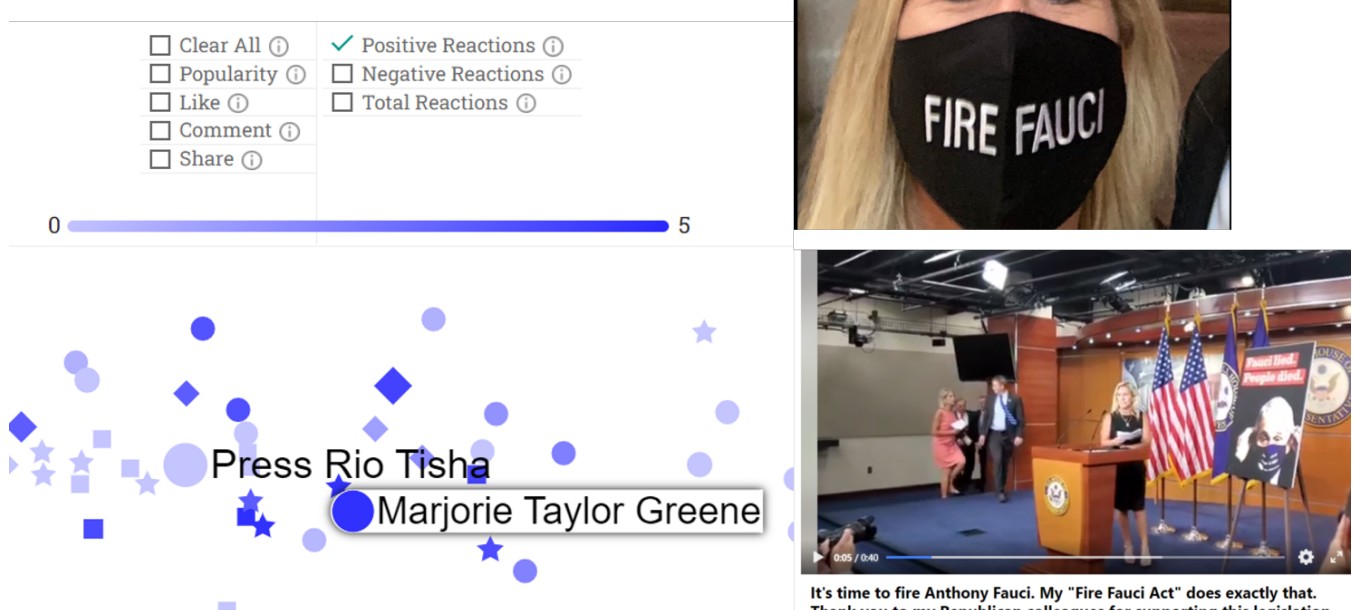

**Figure 10.** Inspecting the positive reactions filter feature and retrieving the popular account of Marjorie Taylor Greene, which aims to influence the public against Fauci.

The tool's search bar allows users to cross-check whether the same user appears in Twitter and Facebook graphs. Although Marjorie Taylor Greene is significantly involved in information dissemination in the Facebook graph, her account does not appear in the Twitter graph. However, Senator Rand Raul, the most famous account in the Twitter graph, is also active in the Facebook graph. Specifically, an image post with text "I told you so" was shared, and it gained a lot of engagement (approx. 2000 comments and 8000 shares).

### 5.1.3. Telegram Analysis

For the analyses of the #FauciGate and #FireFauci use cases, we filtered the collected Telegram data with a time interval of between 1 June 2021 and 15 July 2021. We discovered 985 nodes and 1043 edges, which formulated a graph with 15 communities. The largest community consisted of 381 nodes, while there were nine communities with one node. The word clouds of the largest communities in the graph are illustrated in Figure 11. Apart from Community 6, the rest of the communities' word clouds showed a wide range of topics discussed within the Telegram channels and no specific focus on the Fauci case. This made it more challenging to collect data from Telegram and analyze a particular topic of interest. From the propagation flow feature, it could be seen that half of the top 10 URLs in the graph pointed to Telegram messages or accounts, indicating that the discussions in the Telegram channels stayed within Telegram. The top URL was a YouTube link of a video that had already been removed at the time of the analysis because it violated YouTube's community guidelines. This YouTube video appeared in Community 6, in which @RealMarjorieGreene was the most interactive node. @RealMarjorieGreene is an account that seemed to be involved in the Fauci case during the analyses of the Twitter and Facebook graphs.

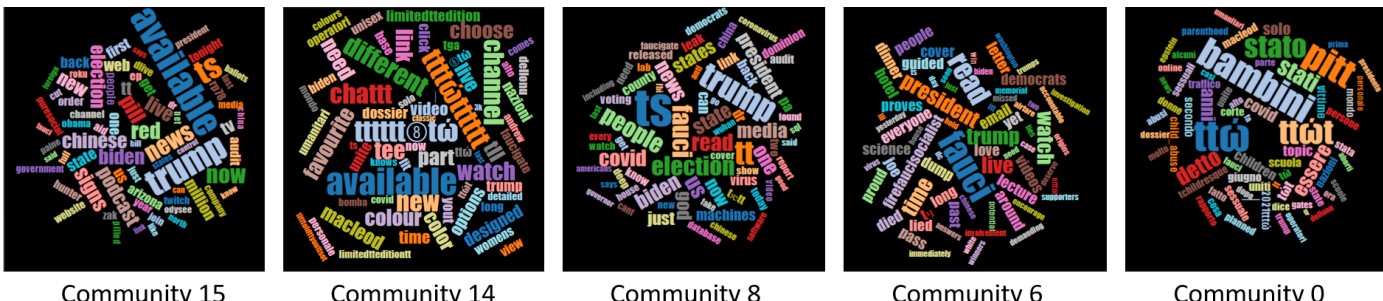

| Community 15 | Community 14 | Community 8 | Community 6 | Community 0 |

**Figure 11.** Telegram word clouds.

We concluded that the analysis and visualization of narratives discussed within Telegram are currently limited. To provide helpful insights and highlight suspicious accounts or behaviors, we need to focus on the data collection step—the limited data acquired by Telegram results in shallow analyses.

*5.2. Hydroxychloroquine*

A similar procedure was followed for the analysis of the hydroxychloroquine use case. The tool detected 110 out of 7078 accounts as suspicious and likely propagating disinformation; 15 of these 110 accounts do not exist on Twitter anymore.

A suspicious account was found in Community 20 with the name *@nyp64N5uCEe3wiu*; it was suspended from Twitter for violating Twitter's rules and policies. Within the network, the account interacted with 27 other accounts in total, and 26 of these interactions replied to the tweets of other accounts or mentioned other accounts. The account acted like a spammer by posting the same text—only hashtags—which was highlighted by the similarity of tweets feature. The hashtags that the account was disseminating included #CCPVirus, #TakeDowntheCCP, #DrLiMengYan, #Hydroxychloroquine, #GTV, #GNews, #NFSC, #WhistleBlowerMovement, #LUDEMedia, #UnrestrictedBioweapon, #COVIDVaccine, #COVID19, #IndiaFightsCOVID, #OriginofCOVID19, #Coronavirus, #WuhanLab, #CCP_is_Terrorist, #CCPisNotChinese, #CCPLiedPeopleDied, and #MilesGuo. Within Community 20, there were also eight accounts that were highlighted as suspicious creating, a doubt about Community 20 as a whole. A user could further investigate each of these accounts individually and draw some conclusions about their participation (or not) in the dissemination of information/disinformation.

*5.3. Big Pharma Use Case*

The tool labeled 220 out of 18,175 accounts as suspicious for the Big Pharma analysis. In all, 54 of these 220 suspicious accounts do not exist on Twitter anymore. An example of a suspicious account was *@UnRapporteur1* (Figure 12), which was the most frequently interacting account of Community 24 (visualized as the larger circle). The account posted the exact same text in its tweets by referencing other accounts. Figure 12 presents the text of the tweets, which contained offensive language. This account is still active on Twitter and is likely suspicious with high values for four features out of nine: the ratio of out-degree to in-degree, the average number of mentions per post and word, and the similarity of tweets' text. Botometer rated this account as a bot with high confidence (5/5).

Another suspicious account that was worth investigating was *@checrai71*, a member of Community 2. The tool flagged this account with the ratio of out-degree to in-degree, the average number of mentions per post and per word, and the followings to followers ratio. The account posted 29 tweets mentioning and replying to 74 different accounts. This account is still active on Twitter and has proven to be a strong supporter of the Big Pharma conspiracy theory based on a video (https://vimeo.com/500025377 accessed on 8 April 2022) shared in its tweets supporting this theory. The Botometer's score of this account was 2.2 (i.e., leaning more toward a "regular user" rather than bot).

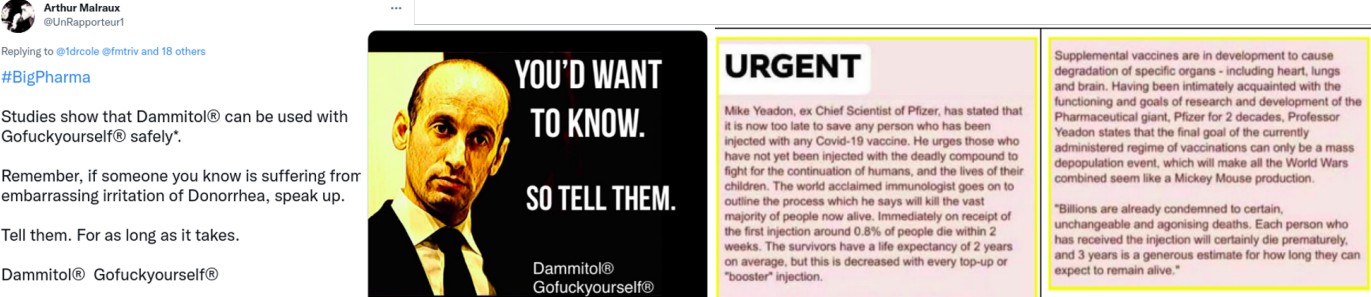

**Figure 12.** Example of a suspicious account in the Big Pharma analysis (**right**) and example of a suspicious account posting conspiracy tweets in the great reset analysis (**left**).

### 5.4. Great Reset Use Case

Regarding the great-reset-related tweets, the tool highlighted 231 out of 14,716 accounts as suspicious. From the highlighted accounts, we found that 29 are not available on Twitter anymore, and we further inspected one of them, which was *@riesdejager*. This account tweeted the conspiracy message presented in Figure 12. It was labeled as having two suspicious features: similarity of tweet texts and average number of URLs per word.

The account with the highest number of suspicious features was *@inbusiness4good*. It was highlighted due to its ratio of out-degree to in-degree, average number of mentions per post, similarity of tweets texts, average number of URLs per word, and average number of hashtags per word. It supported an action called Clim8 Savers, and the purpose was to persuade people to plant trees. The Botometer's score of this account was 3.4.

### 5.5. Quantitative Analysis of Suspicious Users

We further analyzed the developed features that highlighted the users as suspicious and investigated the importance of each feature in the four use cases. As presented in Table 9, the cosine similarity of tweet feature dominated in the hydro and Big Pharma cases. In contrast, the average mentions per post feature was the top feature in the Fauci and great reset cases. Notably, there were cases in which a feature was not involved but dominated in other cases. For example, the average mentions per word feature did not highlight any user in the hydro and great reset cases; however, in the Big Pharma case, this feature labeled 50 out of the 220 likely suspicious users (22.7%). A manual inspection of the four COVID-19-related use cases concluded that it was more likely a user was spreading disinformation when more features highlighted the user. However, there were cases in which even one feature was a solid indication for a further investigation of the user.

**Table 9.** The number of users that each feature highlighted as a suspicious per use case.

| Suspicious Account Features | Fauci | Hydro | Big Pharma | Great Reset |
| --- | --- | --- | --- | --- |
| Out-degree to in-degree | 62 | 3 | 18 | 10 |
| Followings to followers | 28 | 7 | 18 | 13 |
| Following rate | 27 | 14 | 33 | 35 |
| Status rate | 19 | 6 | 16 | 16 |
| Average mentions per post | **143** | 0 | 61 | **110** |
| Average mentions per word | 8 | 0 | 50 | 0 |
| Average hashtags per word | 18 | 37 | 1 | 4 |
| Average URLs per word | 57 | 9 | 11 | 11 |
| Cosine similarity of tweets | 122 | **44** | **62** | 50 |

Table 10 presents the number of users per investigated case and the percentages of nonexistent, suspended, and suspicious users (as labeled by our tool). Additionally, at the bottom of the table, the percentage of users calculated with bot scores (https://botometer.osome.iu.edu/ accessed on 8 April 2022) higher than three are listed.

It is noticeable that the MeVer NetworkX analysis and visualization tool labeled each user with a low percentage (approx. 1–2%) of the total number of users in a network as suspicious. In that way, the tool provided users with a clue to start an investigation and focus on a few users with one or more features that raised suspicions. Using the Twitter API two months after we collected the datasets, we found that out of the detected suspicious users, 10.1% for the Fauci case, 13.6% for the hydro case, 8.6% for the Big Pharma case, and 12.5% for the great reset case were not available on Twitter anymore. In this way, we can consider the highlighted users as likely correct selections (i.e., users that violated Twitter's policies). However, users who remained active but were still highlighted by the tool could spread disinformation or support misleading claims, but Twitter's policies were not violated. Such an example is the account @UnRapporteur1, described in Section 5.4.

**Table 10.** Quantitative analysis of suspicious users for the four COVID-19-related use cases.

| | Fauci | Hydro | Big Pharma | Great Reset |
|---|---|---|---|---|
| All users | 18,310 | 7078 | 18,175 | 14,716 |
| Nonexistent | 1529 | 477 | 1261 | 1187 |
| Suspended | 868 | 257 | 653 | 610 |
| Suspicious | 425 | 110 | 220 | 231 |
| Percentage of suspicious users in relation to all users | 2.3% | 1.6% | 1.2% | 1.6% |
| Percentage of suspicious users not available on Twitter | 10.1% | 13.6% | 8.6% | 12.5% |
| Percentage of suspicious users not available on Twitter due to being suspended | 6.1% | 6.4% | 5.5% | 5.6% |
| Users with bot scores (not available for unavailable users) | 16,635 | 6548 | 16,755 | 13,396 |
| Percentage of users with bot scores $\geq 4$ | 10.7% | 6.8% | 7.0% | 7.12% |
| Percentage of users with 4 >bot scores $\geq 3$ | 13.9% | 21.2% | 19.3% | 16.1% |

## 6. Execution Time

To study the computational behavior of the proposed tool, we collected a large Twitter dataset by querying with the hashtag #COVIDVaccine. The hashtag was selected as it was a trending topic, which could result in large networks. We generated graphs of progressively larger sizes (sums of nodes and edges), starting from ∼1000 and reaching up to ∼140,000, which we considered sufficient for the support of several real-world investigations. We carried out a graph analysis and visualization on a Linux machine with a 2 Intel Xeon E5-2620 v2 and 128 GB of RAMand calculated the time needed for each graph. Figure 13 illustrates the execution times in seconds.

We noticed that for the small graphs (fewer than 10,000 nodes and edges), the execution time increased linearly by a very small factor in relation to the graphs' sizes. For instance, doubling the number of nodes and edges from 700 to 1400 nodes and edges resulted in an increase of 12.5% in execution time. For larger graphs, the execution time increased with a much higher linear factor or even in a super-linear manner. The time needed to analyze and visualize a graph with a size of 140,000 nodes and edges was twice what it took to build a graph with a size of ∼72,000 nodes and edges.

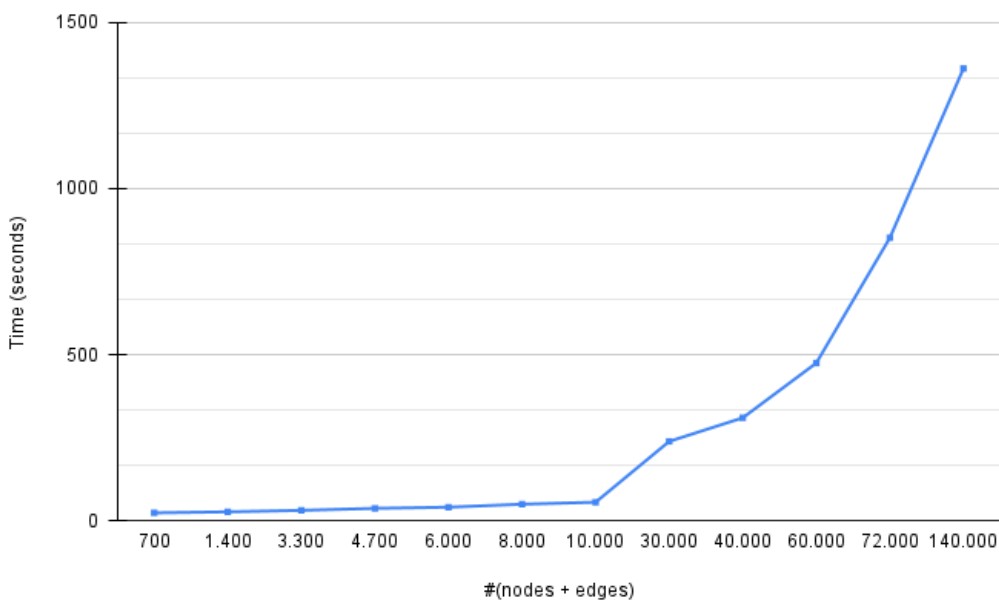

**Figure 13.** Execution times in seconds needed to build and visualize small and large graphs.

## 7. Comparison with Gephi and Hoaxy

**Gephi** is one of the top visualization tools in social network analysis [30,31]. It is a free open-source standalone software used for visualizations and explorations of all kinds of networks. Its advantages include high quality visualizations and the fact that knowledge of programming languages and the ability to handle large graphs are not required. Although Gephi is a leading tool for network visualizations, our MeVer NetworkX analysis and visualization tool is not comparable to Gephi. Gephi is a standalone software, as aforementioned, while the MeVer NetworkX analysis and visualization tool is provided as a web-based application. Gephi provides visualizations for large networks (i.e., it can render networks up to 300,000 nodes and 1,000,000 edges), while our tool supports smaller graphs, focusing on a specific narratives of disinformation. The main difference between the tools that makes them incomparable is that Gephi provides a multitude of functionalities for visualization and filtering, while our tool focuses more on the analysis of the accounts involved in a network, their characteristics, and the information that is disseminated through and among them.

**Hoaxy** is a tool that visualizes the spread of information on Twitter. It supports the uploading of a CSV or JSON file containing Twitter data. For a comparison, we created CSV files compatible with Hoaxy containing the tweets of the four COVID-19-related use cases that we investigated. We submitted each file and created the graphs with Hoaxy. First, we examined the execution time needed to analyze and build the graphs. In Table 11, Hoaxy seems much faster than MeVer NetworkX in all use cases. However, we needed to consider each tool's features to decide which one is faster. Based on this, we created Table 12, in which the features of the two tools are presented side by side. Hoaxy provides far fewer features than the proposed tool. Concerning execution time, Hoaxy is faster in terms of the time needed to analyze input data and build a graph.

**Table 11.** Execution times of MeVer NetworkX analysis and visuazalion tool vs. Hoaxy for the four COVID-19-related use cases.

|  | Fauci | Hydro | Big Pharma | Great Reset |
|---|---|---|---|---|
| Hoaxy (time in s) | 240 | 105 | 230 | 185 |
| MeVer (time in s) | 355 | 139 | 349 | 261 |

Apart from the multitude of features that the MeVer NetworkX tool provides over Hoaxy and its comparable execution time, a significant advantage of the MeVer NetworkX tool is the improved layout with non-overlapping graphs, providing users with an easy-to-digest visualization of communities. Figure 14 illustrates the graphs around the hydro topic in MeVer NetworkX (right) and Hoaxy (left). In the graph built by Hoaxy, the nodes and edges overlap and a user must zoom in and out to investigate them. Instead, the MeVer NetworkX tool provides a simple and clear visualization with different colors among communities. Moreover, the node with its interactions is highlighted by clicking on a node while the rest become blurred. In this way, a user can more easily inspect the nodes of interest one by one.

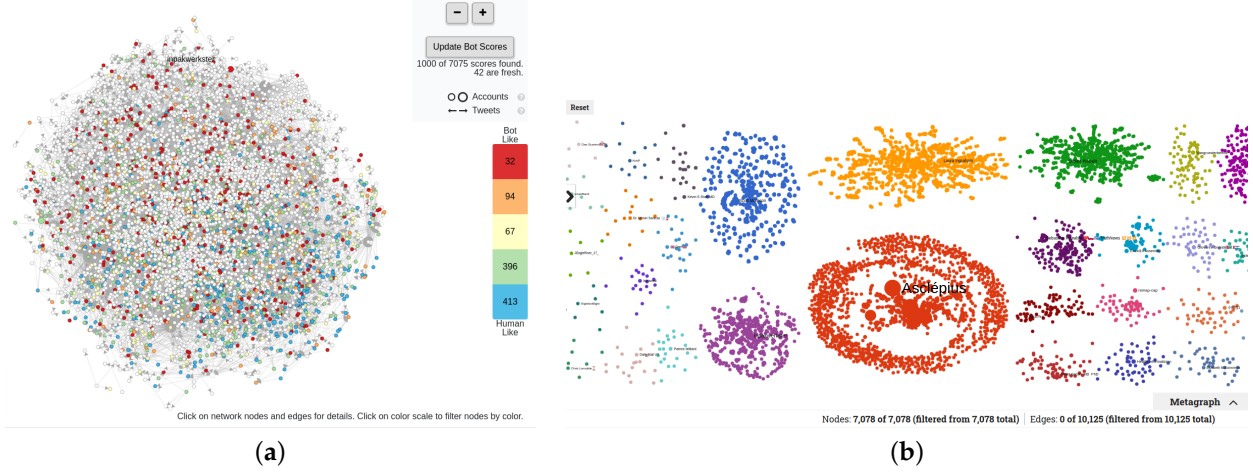

(**a**)                (**b**)

**Figure 14.** Graphs for hydro use case: (**a**) Hoaxy visualization and (**b**) MeVer NetworkX analysis and visualization.

**Table 12.** Comparison of MeVer NetworkX's and Hoaxy's analysis and visualization features.

| Feature | MeVer Tool | Hoaxy | Feature | MeVer Tool | Hoaxy |
|---|---|---|---|---|---|
| Community detection | + | − | Centrality/influence scatter plots | + | − |
| Individual user inspections | + | + | Propagation flows of URLs | + | + |
| Tweet texts | Embedded | External links | Propagation flows of media | + | − |
| Word clouds | + | − | Propagation flows of hashtags | + | − |
| Statistics for each community | + | − | Metagraphs | + | − |
| Hashtag plots | + | − | Tweet timelines | + | + |
| Date plots | + | − | Highlight of suspicious accounts | + | − |

## 8. Discussion and Future Steps

The tool is available upon request ( https://networkx.iti.gr/ accessed on 8 April 2022). To the best of our knowledge, it is the only tool supporting the analysis of multiple platforms and even providing some cross-platform investigations. The tool aims to support the demanding work of journalists and fact checkers to combat disinformation. The advanced functionalities offered by the tool are valuable, as showcased through the presented use cases. The aggregation and visualization capabilities provided to users offer easy ways to navigate large graphs without a need for special knowledge. The developed functionalities offer users a semi-automatic procedure that can increase productivity and save time. The tool's core functionality is to highlight suspicious accounts based on features that are often associated with inauthentic behavior. Although the presented use cases showed that these features are helpful and provide valuable insights about the accounts, in the future, we aim to train models that automatically highlight suspicious users, providing better support to investigators. Additionally, a direction that we are investigating as a future step is integrating third-party tools, such as the Botometer, which provide more insights about

accounts. Finally, a data-collection component is an essential part of the tool. The tool requires GEFX or CSV files containing data collected by a social media platform in question. However, this part of collecting data needs specialized knowledge or some third-party tools. For that reason, we integrated the InVID-WeVerify plugin into the tool in order to offer a smooth and intuitive analysis process and are considering further ways to improve the user experience.

**Author Contributions:** Conceptualization, O.P., T.M. and S.P.; Data curation, O.P., T.M. and F.P.; Formal analysis, O.P. and S.P.; Funding acquisition, S.P. and I.K.; Investigation, O.P.; Methodology, O.P., T.M. and S.P.; Project administration, S.P.; Software, O.P. and L.A.; Supervision, S.P.; Validation, O.P., T.M. and S.P.; Visualization, O.P.; Writing—original draft, O.P. and F.P.; Writing—review & editing, O.P. and S.P. All authors have read and agreed to the published version of the manuscript.

**Funding:** This research was funded by the WeVerify and AI4Media projects, which are funded by the European Commission under contract numbers 825297 and 951911, respectively, as well as the US Paris Tech Challenge Award, which is funded by the US Department of State Global Engagement Center under contract number SGECPD18CA0024.

**Data Availability Statement:** Not Applicable, the study does not report any data.

**Conflicts of Interest:** The authors declare no conflict of interest.

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
