# Peer review of "MeVer NetworkX: Network Analysis and Visualization for Tracing Disinformation"

_futureinternet, doi:10.3390/fi14050147_

Round 1

Reviewer 1 Report

The authors propose a tool to analyse and visualise disinformation networks. The paper is well written and the contribution is interesting, however, I think that the paper can be improved as follows.

  • I have a very hard time understanding what some of the edges mean. For instance, in the case of the Facebook attributes (Table 2): what does it mean that an edge stores the number of likes on a post? What are the edge source and destination nodes in this case? I think the authors should put some effort into describing better these attributes and how they are translated into edges.
  • I also think that the paper lacks a proper description of the methodology used to identify suspect nodes. The authors vaguely hint that the "Average mentions per word" feature may be a sign of suspicious nodes, but I didn't manage to find a proper definition or detection process.
  • While the authors provided an interesting case study for the tool proposed in this paper, I wonder whether the tool is capable of handling large scale graphs or not. In particular, I suggest the authors discuss, fixed a machine with certain computational and storage resources, the limit they achieved using the tools in terms of number of nodes and edges.

Author Response

MeVer NetworkX: Network Analysis and Visualization for Tracing Disinformation: Response to reviewers

We would like to thank the reviewers for their insightful comments. We considerably revised the submission in line with their recommendations. More specifically, we revised the paper in the following aspects: a) we explained in more detailed way parts that were not clear enough such as the meaning of edges and the user of the features for highlighting suspicious users, b) we ran a set of experiments and reported the execution times needed to build and visualize the graph, c) we extended the literature review to illustrate the central topic in a more detailed way, d) we provided a more complete conclusion discussing the findings, strengths and limitations of the present project and finally e) we carried out an additional round of careful proofreading.

Reviewer Comments to Author:

Reviewer: 1

The authors propose a tool to analyse and visualise disinformation networks. The paper is well written and the contribution is interesting, however, I think that the paper can be improved as follows.

  1. I have a very hard time understanding what some of the edges mean. For instance, in the case of the Facebook attributes (Table 2): what does it mean that an edge stores the number of likes on a post? What are the edge source and destination nodes in this case? I think the authors should put some effort into describing better these attributes and how they are translated into edges. Authors’ response: An edge is an interaction among two nodes e.g. when an account shares a URL then an edge is created between these nodes (account, URL). Metadata about the post containing the URL is used as edge’s attributes since the post might be shared by multiple accounts and have multiple edges (interactions). In the revised version, details about the edges’ attributes are included in Section 3.1.
  2. I also think that the paper lacks a proper description of the methodology used to identify suspect nodes. The authors vaguely hint that the "Average mentions per word" feature may be a sign of suspicious nodes, but I didn't manage to find a proper definition or detection process. Authors’ response: The features are inspired by machine learning-based approaches that use these features to train models that detect spam posts and/or bots. The features individually might not stand for a strong indication that the account is suspicious but provide a clue that the account is worth further investigation. The features are provided individually but also in combination. This is a semi-automatic identification where the tool highlights a number of accounts while the user is responsible to investigate them in order to conclude the reliability of the accounts. To address this concern, we modified the text of Section 3.2 paragraph ‘Highlight suspicious accounts.’ and provide more details. 
  3. While the authors provided an interesting case study for the tool proposed in this paper, I wonder whether the tool is capable of handling large scale graphs or not. In particular, I suggest the authors discuss, fixed a machine with certain computational and storage resources, the limit they achieved using the tools in terms of number of nodes and edges.  Authors’ response: We executed a set of experiments with different numbers of nodes and edges (from a small graph of 1000 nodes and edges to a large graph of 140,000 nodes and edges). A Figure presenting the execution times and a discussion of the results is included in section 6 of the revised manuscript.

Reviewer 2 Report

The authors should ask for the help of native English-speaking proofreader, because there are some minor linguistic mistakes that should be fixed.

The literature review should be extended in order to illustrate the central topic in a more detailed way.

It is recommended to include a more extended discussion of the findings, strengths and limitations of the present project with additional explanation/details and also a brief conclusion with future work.

Author Response

MeVer NetworkX: Network Analysis and Visualization for Tracing Disinformation: Response to reviewers

We would like to thank the reviewers for their insightful comments. We considerably revised the submission in line with their recommendations. More specifically, we revised the paper in the following aspects: a) we explained in more detailed way parts that were not clear enough such as the meaning of edges and the user of the features for highlighting suspicious users, b) we ran a set of experiments and reported the execution times needed to build and visualize the graph, c) we extended the literature review to illustrate the central topic in a more detailed way, d) we provided a more complete conclusion discussing the findings, strengths and limitations of the present project and finally e) we carried out an additional round of careful proofreading.

Reviewer Comments to Author:

Reviewer: 2

  1. The authors should ask for the help of native English-speaking proofreader, because there are some minor linguistic mistakes that should be fixed. Authors’ response: We carried out an additional round of careful proofreading. 
  2. The literature review should be extended in order to illustrate the central topic in a more detailed way. Authors’ response: We discuss the three main parts of the tool. First, the data collection part is a challenging step where the user needs to have specialized knowledge to collect data. We provide a brief discussion since this part is not a central part of this work. Second is the bot/spammer detection that we investigated to implement the features/filters that provide clues about suspicious accounts and behaviours in the graph. We provide an overview of the existing approaches and features. Finally, we discuss existing visualization tools which is the core field of this work. In the revised version, we extended the related work coverage, with a special focus on section 2.3, which is the most pertinent for the paper. 
  3. It is recommended to include a more extended discussion of the findings, strengths and limitations of the present project with additional explanation/details and also a brief conclusion with future work. Authors’ response: We extended the Discussion and Future steps section with more details.